# Using whole-genome sequence data to examine the epidemiology of antimicrobial resistance in *Escherichia coli* from wild meso-mammals and environmental sources on swine farms, conservation areas, and the Grand River watershed in southern Ontario, Canada

**Nadine A. Vogt**[1]*, **Benjamin M. Hetman**[1], **Adam A. Vogt**[2], **David L. Pearl**[1], **Richard J. Reid-Smith**[1,3], **E. Jane Parmley**[1], **Stefanie Kadykalo**[3], **Kim Ziebell**[4], **Amrita Bharat**[5,6], **Michael R. Mulvey**[5,6], **Nicol Janecko**[7], **Nicole Ricker**[8], **Samantha E. Allen**[9,10], **Kristin J. Bondo**[8¤], **Claire M. Jardine**[8,11]

**1** Department of Population Medicine, Ontario Veterinary College, University of Guelph, Guelph, Ontario, Canada, **2** Independent Researcher, Mississauga, Ontario, Canada, **3** Centre for Foodborne, Environmental and Zoonotic Infectious Diseases, Public Health Agency of Canada, Guelph, Ontario, Canada, **4** National Microbiology Laboratory, Public Health Agency of Canada, Guelph, Ontario, Canada, **5** National Microbiology Laboratory, Public Health Agency of Canada, Winnipeg, Manitoba, Canada, **6** Department of Medical Microbiology and Infectious Diseases, University of Manitoba, Winnipeg, Manitoba, Canada, **7** Quadram Institute Bioscience, Norwich, United Kingdom, **8** Department of Pathobiology, Ontario Veterinary College, University of Guelph, Guelph, Ontario, Canada, **9** Wyoming Game and Fish Department, Laramie, Wyoming, United States of America, **10** Department of Veterinary Sciences, University of Wyoming, Laramie, Wyoming, United States of America, **11** Canadian Wildlife Health Cooperative, Ontario Veterinary College, Guelph, Ontario, Canada

¤ Current address: Pennsylvania Cooperative Fish and Wildlife Unit, The Pennsylvania State University, University Park, State College, Pennsylvania, United States of America
* nvogt@uoguelph.ca

## Abstract

Antimicrobial resistance (AMR) threatens the health of humans and animals and has repeatedly been detected in wild animal species across the world. This cross-sectional study integrates whole-genome sequence data from *Escherichia coli* isolates with demonstrated phenotypic resistance that originated from a previous longitudinal wildlife study in southern Ontario, as well as phenotypically resistant *E. coli* water isolates previously collected as part of a public health surveillance program. The objective of this work was to assess for evidence of possible transmission of antimicrobial resistance determinants between wild meso-mammals, swine manure pits, and environmental sources on a broad scale in the Grand River watershed, and at a local scale—for the subset of samples collected on both swine farms and conservation areas in the previous wildlife study. Logistic regression models were used to assess potential associations between sampling source, location type (swine farm vs. conservation area), and the occurrence of select resistance genes and predicted plasmids. In total, 200 isolates from the following sources were included: water (n =

**Data Availability Statement:** Data files for isolates from the previous wildlife study are available from the Scholars Portal Dataverse, VI: http://dx.doi.org/10.5683/SP/BLFRK5. All sequence data for these isolates from the previous wildlife study have been deposited to Genbank, and is available under BioProjects PRJNA745182, and PRJNA765107. A list of accession numbers for these isolates is available in Supporting Information. Data for water-derived isolates can only be requested from FoodNet Canada; interested researchers who would like to access these data will need to meet FoodNet Canada's requirements and make a data sharing agreement with them, as the authors of the study did. To request access to FoodNet Canada data, interested researchers can contact: Frank Pollari, DVM, MPH, DVSc. Lead-FoodNet Canada. frank.pollari@phac-aspc.gc.ca.

**Funding:** Funding was provided by the Ontario Ministry of Agriculture, Food, and Rural Affairs (OMAFRA), through the Ontario Agri-Food Innovation Alliance (UofG2016-2642). NAV received stipend funding through the Ontario Veterinary College, the University of Guelph, and a National Sciences and Engineering Research Council Postgraduate Scholarship-Doctoral. The funders had no role in study design, data collection and analysis, decision to publish, or preparation of the manuscript.

**Competing interests:** The authors have declared that no competing interests exist.

20), wildlife (n = 73), swine manure pit (n = 31), soil (n = 73), and dumpsters (n = 3). Several genes and plasmid incompatibility types were significantly more likely to be identified on swine farms compared to conservation areas. Conversely, internationally distributed sequence types (e.g., ST131), extended-spectrum beta-lactamase- and AmpC-producing *E. coli* were isolated in lower prevalences (<10%) and were almost exclusively identified in water sources, or in raccoon and soil isolates obtained from conservation areas. Differences in the odds of detecting resistance genes and predicted plasmids among various sources and location types suggest different primary sources for individual AMR determinants, but, broadly, our findings suggest that raccoons, skunks and opossums in this region may be exposed to AMR pollution via water and agricultural sources, as well as anthropogenic sources in conservation areas.

## Introduction

The emergence and persistence of antimicrobial resistance (AMR) is a major challenge for human health worldwide [1]. Antimicrobial resistant infections directly result in increased morbidity and mortality, and also represent a substantial burden to health care in terms of both cost and efficacy [2, 3]. There is growing evidence to suggest that wild animals play a role in acquiring and disseminating AMR determinants within the environment [4, 5]. Exploratory work, primarily in the form of cross-sectional surveys, has demonstrated that avian and mammalian wildlife may carry a variety of zoonotic agents (e.g., *Salmonella*, *Campylobacter*), some of which have been shown to be resistant to antimicrobials considered critical to human health [6–8]. In addition, bacterial clones of international importance (e.g., *E. coli* ST131), extended spectrum beta-lactamase-producers (ESBLs), and organisms resistant to last-resort antimicrobials (e.g., colistin, vancomycin, carbepenems) mediated by mobile resistance genes have all been isolated from and documented [9–16]. Much of the work examining the epidemiology of AMR in wildlife has been focused on wild birds [4, 17, 18]. However, there is mounting evidence that mammalian wildlife, such as raccoons (*Procyon lotor*), striped skunks (*Mephitis mephitis*), and Virginia opossums (*Didelphis virginiana*), can also harbour antimicrobial resistant bacteria and might therefore represent a potential source of resistant clones or AMR genes for humans and domestic animals [19–25]. With the decreasing cost of whole-genome sequencing, this technology is increasingly being used in wildlife research to simultaneously identify strains, resistance genes, plasmids, and other genetic markers, such as virulence genes, using a single laboratory processing method [26, 27].

Findings from a previous three-year longitudinal study of raccoons and environmental samples in southern Ontario demonstrated that the overall prevalence of AMR among untyped *Escherichia coli* isolates from raccoon fecal samples did not differ significantly between swine farms and conservation areas [19]. However, a comparison of resistance phenotypes and genotypes (determined using PCR) of the resistant *E. coli* isolates in these different location types revealed similar phenotypes and resistance genes among isolates obtained in conservation areas that were altogether absent from isolates obtained on swine farms (e.g., *bla*CMY-2) [19], suggesting that there may, indeed, be differences in the types of AMR determinants carried by raccoons, depending on the local environment. Our recent epidemiologic assessment of the subset of *Salmonella* and *E. coli* isolates originating from swine farms using genomic data revealed frequent overlap, and thus, possible transmission, of AMR determinants between soil and raccoons, but there was limited overlap between isolates from raccoons and swine manure pits [28].

The aim of the present work was to build on the previous longitudinal study [19], and small-scale analysis of genomic data from swine farm isolates [28], by incorporating additional whole-genome sequencing data from *E. coli* isolates obtained in conservation areas, to explore the potential impact of different location types (swine farms vs. conservation areas) on the occurrence of AMR determinants in raccoons. In addition, we sought to examine phenotypically resistant *E. coli* water isolates obtained by routine public health surveillance in the same study region and time period alongside the isolates from the previous wildlife study, to better understand the potential role of raccoons and other meso-mammals in the ecology of AMR in a broader context. Thus, our specific objectives were: 1) to assess for evidence of possible transmission of AMR determinants between wildlife, swine manure pits, and environmental sources at a broad scale in the Grand River watershed, and 2) to assess for evidence of possible transmission of AMR determinants at a local scale, for the subset of wildlife and environmental samples collected in different location types (swine farms vs. conservation areas). Possible transmission of *E. coli* and AMR determinants between different sampling sources was assessed using population structure assessments, and epidemiological modeling of select AMR determinants (i.e., genes, predicted plasmids). The aim of our epidemiological modeling was to infer potential transmission based on the distribution patterns of AMR determinants (determined *in silico*), by assessing the impact of source type and location type (if applicable) on the occurrence of select genes and plasmid incompatibility (Inc) groups. An additional objective was to assess the validity of *in silico* identification of AMR genes, using phenotypic susceptibility test results as the gold standard.

## Methods

### Dataset

*Escherichia coli* isolates examined within this study were obtained from samples collected previously for two different projects/programs. Isolates from wildlife, swine manure pits, and environmental samples, excluding water, were obtained from a previous longitudinal wildlife study on swine farms and conservation areas in southern Ontario (2011–2013) [19, 22]. This previous wildlife study was approved by the Animal Care Committee at the University of Guelph following the guidelines of the Canadian Committee on Animal Care (Permit number: 11R015). Water-derived isolates, collected in the same geographic region and time period as the longitudinal study, were obtained through the FoodNet Canada surveillance program. The study region was located within the Grand River watershed (6800km$^2$), and the surrounding region, which includes Guelph, Kitchener, Waterloo and Cambridge, is a populous region of southern Ontario (~1 million people) that overlaps with intensive agricultural operations and an abundance of natural ecosystems (12 conservation areas; ~19,400 hectares).

**Wildlife, swine manure pit, dumpster, and soil *E. coli* isolates.**   Selection of *E. coli* isolates for sequencing from the previous wildlife study was based on demonstrated phenotypic resistance to at least one of 15 antimicrobials (see details below), as previously reported by Bondo *et al.* [19]. Samples from this previous wildlife study included: paw wipes and fecal swabs from live-trapped raccoons, striped skunks, and Virginia opossums, soil samples, swine manure pit samples (from swine farms only), and dumpster wipes (from conservation areas only). Live trapping was focused on raccoons; however, skunks and opossums that were successfully trapped were also sampled [22]. Methods used to trap and process wildlife have been previously described by Bondo *et al.* [19]. In 2012 only, the paws of captured wildlife were also sampled to assess for surface transmission of microorganisms [29]. Methods used to obtain soil, swine manure pit, and dumpster samples are also available in Bondo *et al.* [19]. In 2011, three *E. coli* isolates were cultured from each sample; for samples with more than one isolate

demonstrating phenotypic resistance, selection of one resistant isolate for sequencing was performed using a random number generator. A total of 203 isolates were available for sequencing from the following sources: dumpster (n = 3), swine manure pit (n = 31), raccoon fecal (n = 53), raccoon paw (n = 16), soil (n = 93), skunk fecal (n = 4), skunk paw (n = 1), opossum fecal (n = 2).

**FoodNet Canada water-derived *E. coli* isolates.** Phenotypically resistant *E. coli* isolates from water samples obtained as part of the FoodNet Canada surveillance program were included in the present study if they were collected in the Region of Waterloo sentinel site; sampling of water for generic *E. coli* in this region was initiated in 2012 and continued through 2013. Water sampling was performed bi-weekly at five core water sites in the Grand River watershed, and three recreational areas in the study region; one of these sites was a conservation area that was also sampled in the wildlife study. Further details regarding water sampling, including a map and description of sites, is available from Kadykalo *et al.* [30]. A total of 20 sequenced isolates from water samples were available for inclusion in this study.

## Previous culture and susceptibility testing

Isolation and susceptibility testing of untyped *E. coli* from water samples and the samples from the previous wildlife study were performed as previously described [19, 30, 31]. Susceptibility to 15 antimicrobials was determined using an automated microbroth dilution system (Sensititre, Thermo Scientific) with the National Antimicrobial Resistance Monitoring System (NARMS) antimicrobial test panel CMV3AGNF: azithromycin (AZM), gentamicin (GEN), kanamycin (KAN), streptomycin (STR), amoxicillin-clavulanic acid (AMC), cefoxitin (FOX), ceftiofur (TIO), ceftriaxone (CRO), ampicillin (AMP), chloramphenicol (CHL), sulfisoxazole (SOX), trimethoprim-sulfamethoxazole (SXT), tetracycline (TCY), nalidixic acid (NAL), and ciprofloxacin (CIP). Isolates with only intermediate resistance were considered susceptible, both in selection for sequencing and for interpretation alongside *in silico* resistance results.

## DNA extraction, whole-genome sequencing and genome assembly

Genomic DNA extractions were performed at the University of Guelph, or at the National Microbiology Laboratory (NML) in Winnipeg, Manitoba. Briefly, cultures of 2 ml broth cultures of *E. coli* were grown overnight, and 1 ml was used in the Qiagen DNEasy plant and tissue 96 kit, using manufacturer protocols (Qiagen, Hilden, Germany). DNA extracts were sequenced at the NML in Guelph, Ontario or at the NML in Winnipeg, Manitoba using Nextera XT libraries and Illumina MiSeq version 3 or NextSeq550 platforms according to manufacturer protocols. Raw reads were assembled using SPAdes [32], as part of the Shovill pipeline (version 1.0.1; https://github.com/tseeman/shovill) with the following settings: "—minlen 200 —mincov 2;—assembler spades;—trim".

## Analysis of whole-genome assemblies

Prediction of legacy multi-locus sequence types was performed using MLST (version 2.19.0; https://github.com/tseemann/mlst), which uses the 7-loci Achtman scheme (https://pubmlst. org/mlst/). Core-genome multi-locus sequence typing (cgMLST) of isolates was performed using the 'fairly simple allele calling' tool *fsac* (version 1.2.0; https://github.com/dorbarker/ fsac) and the 2513-loci Enterobase scheme (https://enterobase.warwick.ac.uk/). Isolates with 30 or more missing loci were considered poor quality and were excluded from any further analyses. Minimum spanning trees generated by the standalone GrapeTree software package (version 1.5) [33] were used to visualize population structure, using the "MSTreeV2" algorithm. A cluster threshold of $k = 50$ was used for all visualizations. This lenient clustering

threshold was selected to provide a general, qualitative assessment of overlap between isolates from different sources, since the threshold of $k = 10$ is used by PulseNet as a starting point to consider isolates as potentially belonging to the same strain [34].

Identification of acquired resistance genes was performed using Abricate (version 0.8.13; https://github.com/tseemann/abricate) and the Resfinder database (updated May–17 2020); settings used were 90% identity and 60% coverage. For the identification of acquired beta-lactamases, the identity and coverage settings were increased to 100% identity and 90% coverage. Sensitivity and specificity of *in silico* AMR prediction were calculated for each antimicrobial class, and overall (i.e., all individual test results pooled); phenotypic test results were considered as the gold standard, and resistance was considered a positive test result. Test sensitivity and specificity were not assessed for drug classes for which chromosomal mutations are responsible for a considerable proportion of expressed resistance (i.e., quinolones) [35]. As a quality control measure, isolates with missing genotypes for resistant phenotypic test results for three or more of the seven antimicrobial classes assessed were examined and excluded from further analyses if the number of missing cgMLST loci exceeded 20. Plasmid Inc types were identified using Abricate (version 0.8.13; https://github.com/tseemann/abricate) and the Plasmidfinder database (updated May–17 2020). Settings used were 98% identity and 70% coverage. Serotyping was performed using ECTyper (version 1.0.0, database version 1.0; https://github.com/phac-nml/ecoli_serotyping) with default settings.

## Statistical analyses

Multi-level logistic regression was used to model the odds of select plasmid Inc types and AMR genes found in *E. coli* from different sources. All statistical tests were performed using STATA (STATA Intercooled 14.2; StataCorp, College Station, Texas, USA). Only Inc types and resistance genes present in at least 10% and less than 90% of isolates were modeled statistically. In addition to examination of the overall dataset (A), analyses were also performed for a subset of the data (B) consisting of soil and wildlife isolates collected from conservation areas and swine farms (Fig 1). Source type was examined as an independent variable for both datasets (A) and (B), and the categorization of sampling sources for each dataset is provided in Fig 1. For the subset of data (B), location type (swine farm vs. conservation area) was also examined as an independent variable for sources replicated in both location types (i.e., soil and wildlife isolates). For the full dataset (A), we did not evaluate location type as an independent variable since water isolates originated from a variety of location types across the watershed that could not be characterized by a single group distinct from swine farms and conservation areas. A causal diagram illustrating the relationships between the different variables is provided in Fig 2. The mixed logistic regression models included random intercepts to account for clustering at the site-level, and at the level of the individual sampling source (i.e., animal, swine manure pit, dumpster). Models that did not converge using the 'melogit' command were fitted using 'meqrlogit', which uses QR decomposition methods. Sampling year was included in these mixed models if it confounded the association between source or location type (i.e., its addition caused a >20% change in the coefficient of those variables) [36], or if it was statistically significant ($p \leq 0.05$). Model fit was assessed by examining the best linear unbiased predictions for normality and homoscedasticity, and Pearson's residuals were examined for potential outliers. If variance components were very small ($<1 \times 10^{-3}$), the Bayesian Information Criterion (BIC) value was used to compare the fit of the multi-level logistic regression model to an ordinary logistic regression, and the better fitting model was reported [36]. All tests were two-tailed, and a significance level of $\alpha = 0.05$ was used for all analyses.

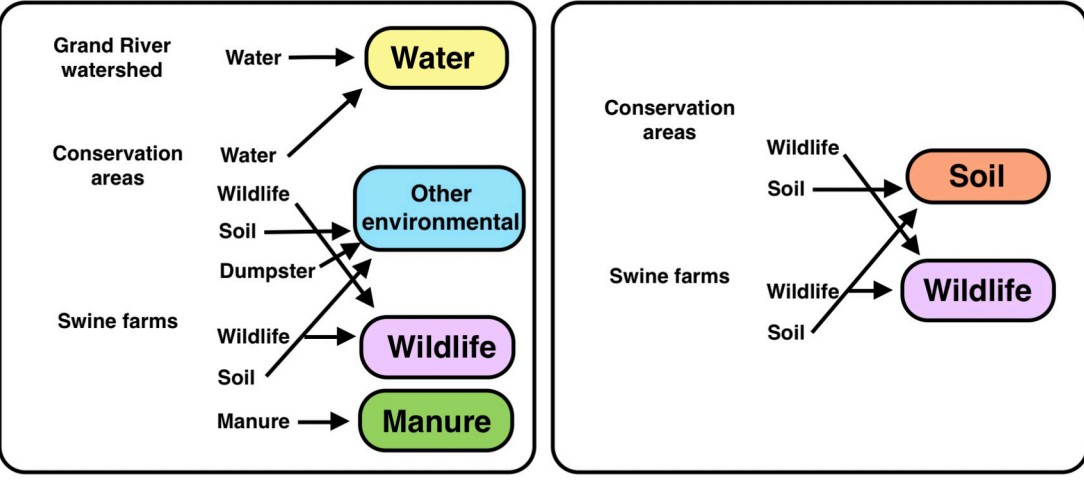

**Fig 1.** Classification of source types for overall dataset (A), and subset of data (B), with sample sizes and independent variables analysed of *Escherichia coli* isolates collected in southern Ontario, 2011–2013.

## Results

### Description of dataset

A total of 223 sequenced isolates were available for inclusion. Following exclusion based on 30 or greater missing loci with the 2513-loci cgMLST scheme, 200 isolates with the following

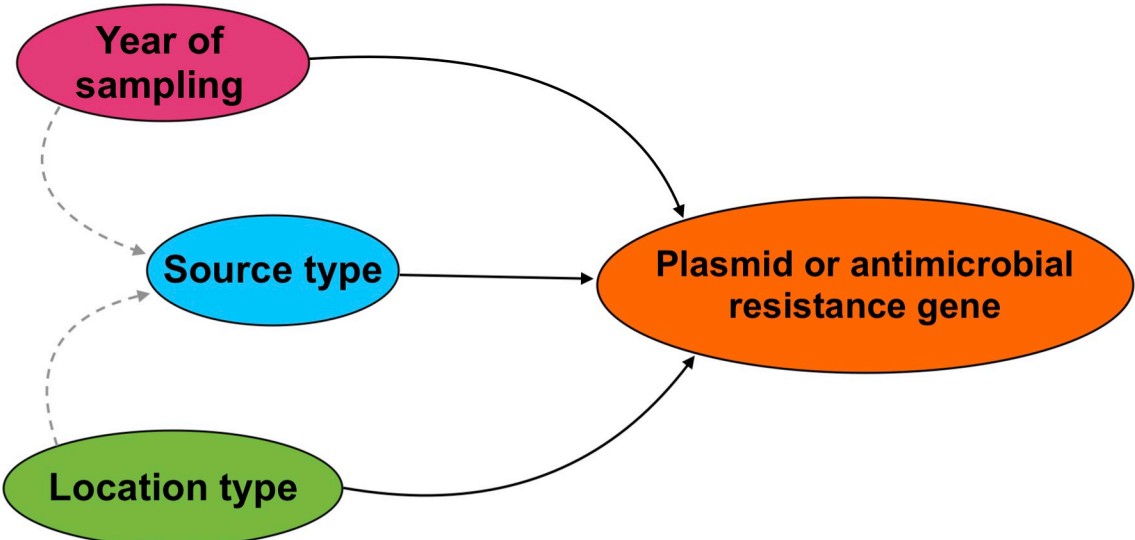

**Fig 2. Causal diagram illustrating the relationships between source type, location type, year of sampling, and the carriage of predicted plasmids and antimicrobial resistance genes in *Escherichia coli* isolates collected from wildlife, swine manure pits, and environmental sources in southern Ontario, 2011–2013.** Solid lines show directionality of proposed relationships between dependent and independent variables. Dashed grey lines show potential confounding relationships.

source distribution were available for subsequent analyses: water (n = 20), swine manure pit (n = 31), wildlife (n = 73), and other environmental sources (n = 76; 73 soil isolates, 3 dumpster isolates). Accession numbers for isolates from the previous wildlife study are available in S1 File. Wildlife isolates originated from 58 unique raccoons, two opossums, and four skunks. In several cases, isolates were obtained from the same animal captured on different occasions; two isolates from one skunk, two isolates from five different raccoons, and four isolates from one raccoon. For the subset of isolates (B) collected from the same source type on both swine farms and conservation areas (i.e., wildlife and soil, n = 146), the following distribution was noted in conservation areas: wildlife (n = 46; 31 raccoon fecal, 12 raccoon paw, 2 opossum fecal, 1 skunk fecal), and soil (n = 28). For the subset of isolates collected on swine farms (n = 103), the distribution was as follows: swine manure pit (n = 31), soil (n = 45), and wildlife (n = 27; 20 raccoon fecal, 3 raccoon paw, 3 skunk fecal, 1 skunk paw). Isolates were obtained from a total of 15 sites in the Grand River watershed: five swine farms, six conservation areas, and four river sites in the region. Isolates were roughly evenly distributed across different years (2011, n = 63; 2012, n = 58; 2013, n = 79), but for isolates collected from water, the majority were collected in 2013 (n = 17/20), and the remaining three isolates were obtained in 2012.

## Distribution of serotypes and MLST types

In total, 113 serovars representing 94 sequence types were identified. Eight isolates were not typeable by MLST (1 water, 2 manure, 1 dumpster, 1 skunk, 1 raccoon, 2 soil), due to a missing allele or a partial match, and 1 isolate could not be serotyped. Serovars consistent with pathogenic *E. coli* strains (e.g., Shiga-toxin producing strains) and sequence types of international importance were identified [37], among which several isolates also demonstrated AMR. Based on identified serotypes, two non-O157 *E. coli* isolates were identified: one O103:H21 ST2354 isolate with phenotypic resistance to streptomycin from a skunk, and one O103:H2 ST2307 isolate with phenotypic resistance to ampicillin, sulfisoxazole, and trimethoprim/sulfamethoxazole from a soil isolate. A number of water, soil and wildlife *E. coli* isolates were also identified as internationally important sequence types responsible for causing urinary tract and bloodstream infections in humans (ST69, ST95, and ST131) [37]; most of these isolates were identified in the Grand River (upstream of the drinking water intake), and in conservation area 1 (Table 1). In addition to ST131, other major sequences types associated with uropathogenic *E. coli* (UPEC) strains were identified in raccoons in conservation areas: fourteen ST10 isolates, and two ST127 isolates [38]. Apart from ST10, none of the isolates from swine manure pits contained these major sequence types (ST69, ST73, ST95, ST96, ST127, ST131 or ST140). A summary of the sequence types and serovars identified are available in S1 and S2 Tables.

## Population structure based on cgMLST

The population structure of *E. coli* for the overall dataset (A), and for subset (B) are presented in Figs 3 and 4, respectively. Similar or identical *E. coli* subtypes were identified from diverse sources (Figs 3A, 4A), regardless of the location type (for wildlife and soil isolates; Fig 4B), or the degree of AMR (Figs 3B, 4C).

## *In silico* determination of AMR genes and plasmid Inc types

In total, 43 resistance genes and 28 plasmid Inc groups were identified (Tables 2 and 3, S3 Table). The most commonly identified resistance genes, with >10% overall prevalence were: $bla_{TEM-1}$, *tet(A)*, *tet(B)*, *sul1*, *sul2*, *aph(6)-Id*, *ant(3")-Ia*, *aph(3")-Ib* (Table 2). Three plasmid types were identified with a >10% overall prevalence: IncFIB(001918), IncI1(1-alpha), and IncFII (Table 3). Resistance genes of human health importance were also identified (e.g.,

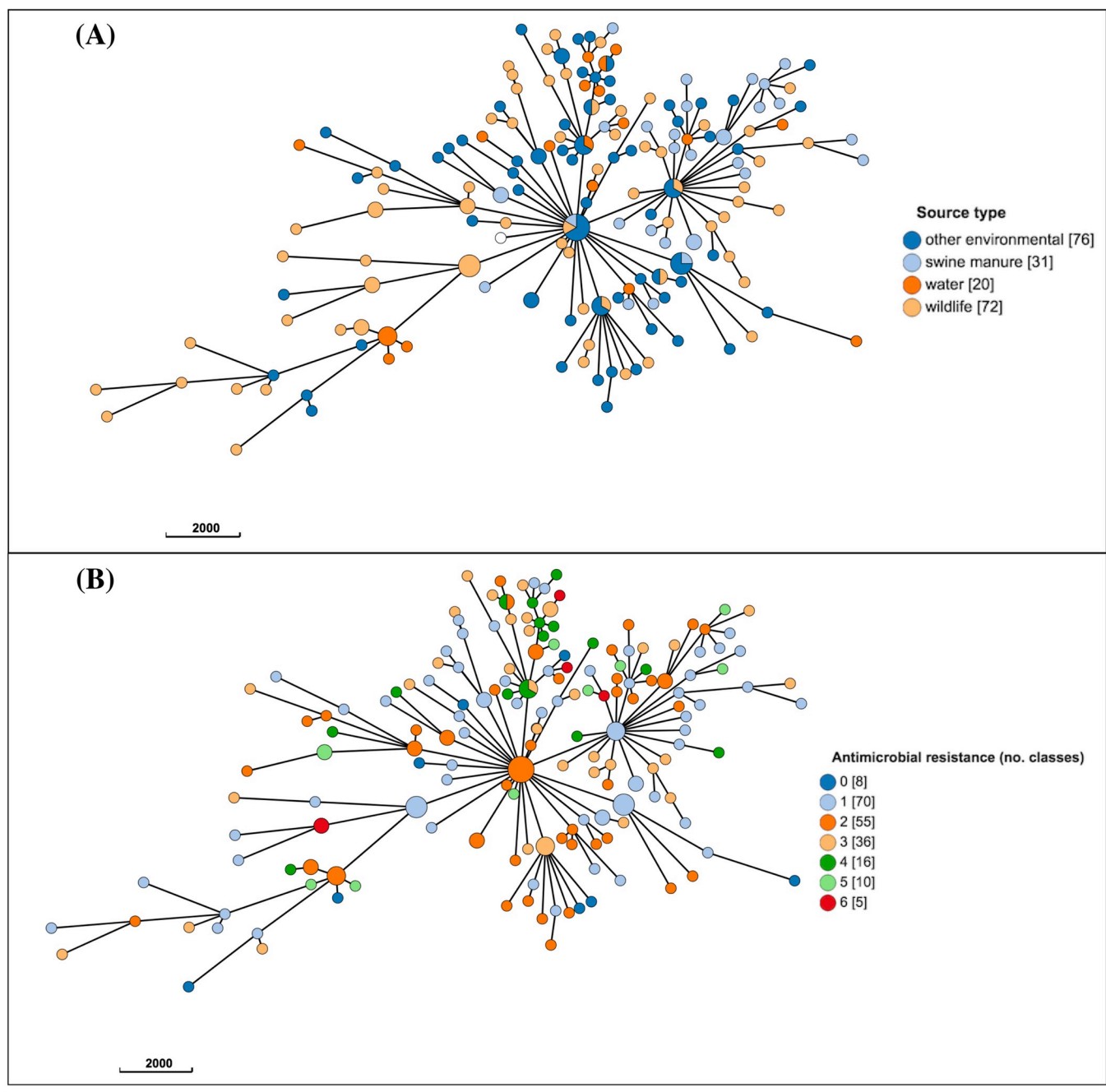

**Fig 3. Population structure of 200 *Escherichia coli* isolates (dataset A) from wildlife, swine manure pits, and environmental sources in southern Ontario based on 2513-loci cgMLST scheme from Enterobase.** Minimum spanning tree created using *k = 50* clustering threshold in *GrapeTree*. (A) Distribution of source types. (B) Antimicrobial resistance by number of drug classes. Frequency counts are in square brackets. Bubble size is proportional to the number of isolates in each cluster, and each cluster contains isolates differing at a maximum of 50 cgMLST loci.

AmpC-producers, ESBLs); ten *E. coli* isolates contained *bla*$_{CMY-2}$, and one raccoon *E. coli* isolate contained a *bla*$_{TEM-35}$. Of the isolates containing *bla*$_{CMY-2}$, two were multi-drug resistant (MDR; 3+ antimicrobial classes) isolates collected from raccoons, and all but two of these isolates were obtained from raccoons captured in conservation areas. One of the two MDR *E. coli* isolates from raccoons had the phenotypic resistance pattern

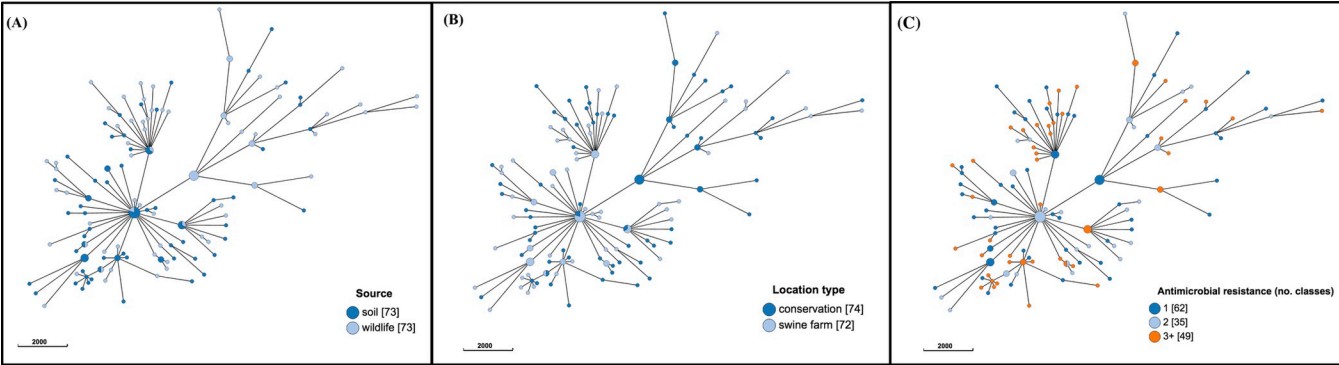

**Fig 4. Population structure of 146 *Escherichia coli* isolates (dataset B) from wildlife and soil on swine farms and conservation areas in southern Ontario based on 2513-loci cgLMST scheme from Enterobase.** Minimum spanning tree created using *k* = 50 clustering threshold in *GrapeTree*. (A) Distribution of sources. (B) Distribution by location type. (C) Antimicrobial resistance by number of drug classes. Frequency counts are in square brackets. Bubble size is proportional to the number of isolates in each cluster, and each cluster contains isolates differing at a maximum of 50 cgMLST loci.

AMC-AMP-FOX-TIO-CRO-CHL-GEN-NAL-STR-SOX-TCY and intermediate resistance to CIP and contained resistance genes *aac(3)-IVa*, *ant(3")-Ia*, *aph(3")-Ib*, *aph(6)-Id*, *floR*, *sul1*, *sul2*, and *tet(A)* in addition to $bla_{\text{CMY-2}}$; this particular isolate also contained two plasmid Inc types, IncC and IncFIBAP001918. The other MDR *E. coli* isolate from a raccoon on a swine farm was resistant to AMC-AMP-FOX-TIO-CRO-CHL-STR-SOX-TCY-SXT, and, in addition to $bla_{\text{CMY-2}}$, contained genes *aadA2*, *aph(3")-Ib*, *dfrA12*, *floR*, *sul1*, *sul2*, *tet(A)*, and an IncC plasmid. An *E. coli* water isolate from the Grand River was identified with a $bla_{\text{OXA-1}}$; this isolate demonstrated phenotypic resistance to AMP-CHL-STR-SOX-TCY, as well as intermediate resistance to AMC, and contained genes *ant(3")-Ia*, *aph(3")-Ib*, *aph(6)-Id*, *floR*, *tet(A)*, *tet(B)*, and an IncX1-1 plasmid. Genes conferring quinolone resistance were also identified: *qnrS1* was identified in an isolate from a raccoon captured in a conservation area with IncI1(1-alpha) and IncX1-1 plasmids, and a *qnrB19* gene was identified in a water isolate that displayed phenotypic resistance to AMP-CHL-GEN-STR-SOX-TCY-SXT and intermediate resistance to CIP, and also contained genes *aac(3)-IId*, *aadA5*, *aph(3")-Ib*, *aph(6)-Id*, *catA1*, *dfrA1*, *sul2*, $bla_{\text{TEM-1}}$, and an IncFIBAP001918 plasmid. Other resistance genes such as $bla_{\text{CARB-2}}$ and *catA1* were identified in less than five isolates in the wildlife subset and were found only in isolates obtained from conservation areas.

## Sensitivity and specificity of *in silico* AMR prediction

The overall sensitivity and specificity of *in silico* identification of resistance genes were 95.9% and 95.5%, respectively (Table 4). Test sensitivity and specificity were at least 95% for all drug classes, except for test sensitivity of beta-lactams (90.2%), and test specificity of aminoglycosides (81.8%).

## Associations between source type, location type and the carriage of resistance genes and plasmid Inc types

Genes *tet(B)*, $bla_{\text{TEM-1}}$, and *sul2* were significantly associated with source type in the overall dataset (Table 5). The odds of identifying *tet(B)* were significantly greater for swine manure pit isolates compared to isolates from wildlife and other environmental sources, but there were no significant differences between other sources (Tables 5 and 6). The odds of identifying *sul2* were greater in isolates from water compared to all other sources, and they were significantly greater in isolates from wildlife and other environmental sources compared to swine manure

**Table 1. Distribution of *Escherichia coli* sequence types of international importance identified in phenotypically resistant isolates collected from wildlife and environmental sources[a] in southern Ontario, Canada 2011–2013 (n = 200).**

| NCBI Accession Number | Isolate Id | Sequence Type | Serotype | Source Type | Resistance Pattern[b] | Resistance Genes | Plasmid Incompatibility Group(s) | Location Type | Location Id |
|---|---|---|---|---|---|---|---|---|---|
| - | - | ST69 | O77/O17/O44/O106:H19 | Water | NAL | None identified | IncB/O/K/Z | Grand River, drinking water intake | 13 |
| - | - | | O77/O17/O44/O106/O73:H18 | Water | AMP-AZM-CIP-NAL-STR-SOX-TCY-SXT | $aph(3")$-$Ib$, $aph(6)$-$Id$, $dfrA14$, $mphA$, $sul2$, $tet(B)$, $bla_{TEM-1}$ | Col156, IncFIA(HI1), IncFIA, IncFII (pRSB107), IncY | Grand River, drinking water intake | 13 |
| - | - | | O25:H4 | Water | KAN-STR-SOX-TCY | $tet(B)$, $bla_{TEM-1}$ | Col156, IncFIB (AP001918) | Grand River, drinking water intake | 13 |
| - | - | | O25:H4 | Water | AMP-NAL-TCY | $tet(B)$, $bla_{TEM-1}$ | Col156, IncFIB (AP001918) | Grand River, drinking water intake | 13 |
| - | - | | O25:H4 | Water | AMP-NAL-TCY | $tet(B)$, $bla_{TEM-1}$ | IncB/O/K/Z, IncFIB (AP001918) | Grand River, drinking water intake | 13 |
| JAIUVT000000000 | N18-00294 | | O77/O17/O44/O106/O73:H18 | Raccoon fecal | AMP-TCY | $tet(A)$, $bla_{TEM-1}$ | IncFII, IncFIB (AP001918) | Conservation area | 4 |
| JAIUUZ000000000 | N18-00347 | | O77/O17/O44/O106:H18 | Raccoon fecal | AMP-KAN-STR-SOX-TCY | $aph(3")$-$Ib$, $aph(3')$-$Ia$, $aph(6)$-$Id$, $sul2$, $tet(B)$, $bla_{TEM-1}$ | IncFIA, IncFIB (AP001918) | Conservation area | 1 |
| JAIUUI000000000 | N18-00377 | | O77/O17/O44/O106:H18 | Raccoon paw | AMP-TCY | $tet(A)$, $bla_{TEM-1}$ | IncFII, IncFIB (AP001918) | Conservation area | 5 |
| JAIUUV000000000 | N18-00354 | ST95 | O25:H4 | Raccoon paw | CHL-SOX-SXT | $aadA2$, $ant(3")$-$Ia$, $cmlA1$, $dfrA12$, $mefB$, $sul3$ | IncFIB(AP001918) | Conservation area | 1 |
| JAIUSZ000000000 | N18-04249 | ST131 | O25:H4 | Raccoon fecal | CIP-NAL-TCY | $tet(A)$, $bla_{TEM-1}$ | IncFIA, IncFII | Conservation area | 1 |
| JAIUVI000000000 | N18-00331 | | O25:H4 | Soil | AMP-TCY | $tet(A)$, $bla_{TEM-1}$ | IncFII, IncFIB (AP001918) | Conservation area | 1 |

[a] No sequence types of international importance were identified in swine manure pit isolates.

[b] AMP = ampicillin; CHL = chloramphenicol; CRO = ceftriaxone; FOX = cefoxitin; SOX = sulfisoxazole; STR = streptomycin; SXT = trimethoprim sulfamethoxazole; TCY = tetracycline; NAL = nalidixic acid; CIP = ciprofloxacin.; KAN = kanamycin.

**Table 2. Frequencies of acquired antimicrobial resistance genes identified using whole-genome sequence data from phenotypically resistant *Escherichia coli* isolates from wildlife, swine manure pits, and environmental sources in southern Ontario, Canada 2011–2013 (n = 200).**

| Antimicrobial Group | Resistance Gene | Accession No.[†] | Wildlife[a] (n = 73) | Swine Manure Pit (n = 31) | Water (n = 20) | Other Environmental[b] (n = 76) | Total (%) |
|---|---|---|---|---|---|---|---|
| Aminoglycoside | *aac(3)-IId* | EU022314 | 3 | 0 | 1 | 1 | 5 (2.5%) |
| | *aac(3)-IVa* | NC_009838 | 4 | 0 | 1 | 3 | 8 (4.0%) |
| | *aadA2* | JQ364967 | 4 | 1 | 1 | 3 | 9 (4.5%) |
| | *aadA5* | AF137361 | 6 | 0 | 1 | 1 | 8 (4.0%) |
| | *ant(3")-Ia* | X02340 | 11 | 7 | 3 | 14 | 35 (17.5%) |
| | *aph(3')-Ia* | V00359/ EF015636 | 5 | 0 | 3 | 4 | 12 (6.0%) |
| | *aph(3')-IIa* | V00618 | 0 | 1 | 0 | 0 | 1 (0.5%) |
| | *aph(3")-Ib* | AF321551/ AF024602 | 24 | 13 | 11 | 29 | 77 (38.5%) |
| | *aph(4)-Ia* | V01499 | 0 | 0 | 1 | 0 | 1 (0.5%) |
| | *aph(6)-Ic* | X01702 | 0 | 1 | 0 | 0 | 1 (0.5%) |
| | *aph(6)-Id* | M28829 | 24 | 13 | 11 | 29 | 77 (38.5%) |
| Beta-lactam | *bla*$_{CMY-2}$ | X91840 | 7 | 0 | 1 | 2 | 10 (5.0%) |
| | *bla*$_{TEM-1}$ | AY458016/ HM749966/ FJ560503 | 28 | 7 | 11 | 17 | 63 (31.5%) |
| | *bla*$_{TEM-35}$ | KP860986 | 1 | 0 | 0 | 0 | 1 (0.5%) |
| | *bla*$_{CARB-2}$ | M69058 | 0 | 0 | 0 | 1 | 1 (0.5%) |
| | *bla*$_{OXA-1}$ | HQ170510 | 0 | 0 | 1 | 0 | 1 (0.5%) |
| Lincosamide | *lnuC* | AY928180 | 0 | 0 | 0 | 1 | 1 (0.5%) |
| | *lnuF* | EU118119 | 0 | 0 | 0 | 1 | 1 (0.5%) |
| Macrolide | *mphA* | U36578 | 3 | 0 | 1 | 0 | 4 (2.0%) |
| | *mphB* | D85892 | 0 | 0 | 1 | 0 | 1 (0.5%) |
| | *mef(B)* | FJ196385 | 1 | 0 | 0 | 1 | 2 (1.0%) |
| | *ereA* | DQ157752 | 0 | 0 | 0 | 1 | 1 (0.5%) |
| Folate pathway inhibitors | *dfrA1* | AF203818/ X00926 | 2 | 1 | 1 | 5 | 9 (4.5%) |
| | *dfrA5* | X12868 | 1 | 1 | 3 | 4 | 9 (4.5%) |

*(Continued)*

**Table 2.** (Continued)

| Antimicrobial Group | Resistance Gene | Accession No.[†] | Wildlife[a] (n = 73) | Swine Manure Pit (n = 31) | Water (n = 20) | Other Environmental[b] (n = 76) | Total (%) |
|---|---|---|---|---|---|---|---|
| | *dfrA8* | U10186 | 1 | 0 | 0 | 0 | 1 (0.5%) |
| | *drfA12* | AM040708 | 3 | 0 | 0 | 1 | 4 (2.0%) |
| | *dfrA14* | DQ388123 | 3 | 0 | 1 | 5 | 9 (4.5%) |
| | *dfrA16* | AF174129 | 0 | 0 | 0 | 1 | 1 (0.5%) |
| | *dfrA17* | FJ460238 | 6 | 0 | 1 | 1 | 8 (4.0%) |
| | *dfrA23* | AJ746361 | 1 | 0 | 0 | 0 | 1 (0.5%) |
| | *sul1* | EU780013 | 14 | 2 | 1 | 8 | 25 (12.5%) |
| | *sul2* | HQ840942/ AY034138 | 18 | 1 | 10 | 16 | 45 (22.5%) |
| | *sul3* | AJ459418 | 3 | 1 | 1 | 3 | 8 (4.0%) |
| Phenicol | *floR* | AF118107 | 7 | 1 | 2 | 4 | 14 (7.0%) |
| | *catA1* | V00622 | 2 | 0 | 2 | 0 | 4 (2.0%) |
| | *cmlA1* | M64556 | 2 | 1 | 1 | 2 | 6 (3.0%) |
| Quinolone | *QnrB19* | EU432277 | 0 | 0 | 1 | 0 | 1 (0.5%) |
| | *QnrS1* | AB187515 | 1 | 0 | 0 | 0 | 1 (0.5%) |
| Fosfomycin | *fosA7* | LAPJ01000014 | 1 | 0 | 0 | 3 | 4 (2.0%) |
| Tetracycline | *tet(A)* | AF534183 | 30 | 11 | 5 | 37 | 83 (41.5%) |
| | *tet(B)* | AF326777/ AP000342 | 19 | 16 | 10 | 19 | 64 (32.0%) |
| | *tet(C)* | AY046276/ AF055345 | 0 | 0 | 0 | 3 | 3 (1.5%) |
| | *tetM* | X04388 | 0 | 0 | 0 | 1 | 1 (0.5%) |

[†] Values from *Resfinder* database.
[a] Includes fecal isolates from raccoons (n = 51), skunks (n = 4), opossums (n = 2), and paw swab samples from raccoons (n = 14), and one skunk.
[b] Includes soil (n = 73) and dumpster isolates (n = 3).

pit isolates (Tables 5 and 6). The odds of identifying $bla_{\text{TEM-1}}$ were significantly greater in water isolates compared to swine manure and other environmental isolates, and higher in wildlife compared to other environmental isolates (Tables 5 and 6). No associations with source type were identified for the remaining genes and plasmid Inc types examined (Table 5).

**Table 3. Frequencies of plasmid incompatibility (Inc) types identified using whole-genome sequence data from phenotypically resistant *Escherichia* isolates obtained from wildlife, swine manure pits, and environmental sources in southern Ontario, Canada, 2011–2013 (n = 200).**

| Inc type | Total (n) | (% of 200) |
|---|---|---|
| IncFIB(AP001918) | 79 | (39.5%) |
| IncI1(alpha) | 32 | (16.0%) |
| IncFII | 29 | (14.5%) |
| IncFIA | 19 | (9.5%) |
| p0111 | 19 | (9.5%) |
| IncY | 17 | (8.5%) |
| IncX1-1 | 12 | (6.0%) |
| IncQ1 | 14 | (7.0%) |
| Col156 | 10 | (5.0%) |

[a] Inc types identified in fewer than 10 isolates included: IncR (n = 9), IncFIA(HI1) (n = 6), IncFIC(FII) (n = 5), IncFIB(K) (n = 4), IncFII(29), (n = 4), IncHI2A (n = 3), IncHI2 (n = 3), IncA/C2 (n = 3), IncB/O/K/Z (n = 2), IncFII (pHN7A8) (n = 2), ColBS512 (n = 2), ColE10 (n = 2), ColpVC (n = 2), IncFIB(pB171) (n = 1), ColIMGS31 (n = 1), IncFII(pRSB107) (n = 1), IncHI1A(CIT) (n = 1), IncHI1B(CIT) (n = 1), IncX1-4 (n = 1).

Among statistically significant models in dataset (A), clustering by sampling site was noted for both the *tet(B)* and $bla_{TEM-1}$ models, and model assumptions were met.

In the subset (B) of isolates that were collected in both location types, only $bla_{TEM-1}$ was significantly associated with source, with the odds of identifying this gene being significantly greater in isolates from wildlife compared to those from soil, and sampling site was retained as a random intercept for this model (Table 7). The following genes and plasmid types had a significantly greater odds of identification in isolates collected on swine farms compared with conservation areas: plasmid type IncFIB(AP001918), *aph(3")-Ib*, *tet(A)*, and *aph(6)-Id* (Table 6). Plasmid types IncFII, and IncI1(1-alpha), and genes *ant(3")-Ia*, *sul1*, *sul2*, and *tet(B)* were not associated with either source type or location type (Table 7). All model assumptions were met.

## Discussion

Previous work examining the role of wildlife in the maintenance and transmission of AMR has often, but not always, shown that wild animals living in close proximity to humans are

**Table 4. Test sensitivity and specificity[a] for *in silico* identification of antimicrobial resistance genes in *Escherichia coli* isolates from wildlife, swine manure pits, and environmental sources in southern Ontario, 2011–2013 (n = 200).**

| Antimicrobial class | Test Sensitivity (95%CI) | Test Specificity (95%CI) |
|---|---|---|
| Aminoglycoside | 95.5% (89–99%) | 81.8% (73–88%) |
| Beta-lactam | 90.2% (82–96%) | 98.3% (94–99%) |
| Macrolide | 100% (16–100%) | 97.0% (93–99%) |
| Sulfonamide | 97.0% (90–97%) | 97.0% (92–99%) |
| Phenicol | 95.6% (78–99%) | 99.4% (97–99%) |
| Tetracycline | 98.6% (95–99%) | 96.2% (87–99%) |
| **Overall[b]** | **95.9% (93–98%)** | **95.5% (94–97%)** |

[a] Phenotypic antimicrobial resistance test results were considered the gold standard. Detection of 15 antimicrobials performed using the CMV3AGNF panel from National Antimicrobial Resistance Monitoring System (Sensititre, Thermo Scientific). *In silico* acquired resistance genes detected using Abricate and the Resfinder database.
[b] Raw counts for all isolates and antimicrobials were pooled together.

**Table 5. Logistic regression models[a,b,c] assessing the association between source type and the occurrence of select plasmid incompatibility types and antimicrobial resistance genes in phenotypically resistant *Escherichia coli* isolates collected from wildlife, swine manure pits, and environmental sources in southern Ontario, 2011–2013 (n = 200, dataset A).**

| | *tet(A)*[a] | | *tet(B)*[b,c] | | *bla*TEM-1[b] | | *sul1*[b] | |
|---|---|---|---|---|---|---|---|---|
| **Source type** | **OR (95% CI)** | ***p*-value** | **OR (95%CI)** | ***p*-value** | **OR (95% CI)** | ***p*-value** | **OR (95% CI)** | ***p*-value** |
| Swine manure pit | REF | **0.216 (global)** | REF | **0.021 (global)** | REF | **0.029 (global)** | REF | **0.161 (global)** |
| Water | 0.61 (0.17 −2.19) | 0.433 | 0.92 (0.26 −3.19) | 0.892 | 4.03 (1.01 −16.18) | 0.050 | 0.72 (0.05 −9.63) | 0.805 |
| Wildlife | 1.27 (0.53 −3.03) | 0.593 | 0.28 (0.11 −0.74) | 0.010 | 2.11 (0.74 −6.01) | 0.161 | 3.76 (0.75 −18.84) | 0.107 |
| Other environmental[d] | 1.72 (0.73 −4.09) | 0.215 | 0.32 (0.13 −0.79) | 0.013 | 0.88 (0.31 −2.54) | 0.815 | 1.75 (0.34 −9.05) | 0.504 |

| | *sul2*[a] | | *ant(3")-Ia*[a] | | *aph(3")-Ib*[b] | | *aph(6)-Id*[b] | |
|---|---|---|---|---|---|---|---|---|
| **Source type** | **OR (95% CI)** | ***p*-value** | **OR (95%CI)** | ***p*-value** | **OR (95% CI)** | ***p*-value** | **OR (95% CI)** | ***p*-value** |
| Swine manure pit | REF | **<0.001 (global)** | REF | **0.810 (global)** | REF | **0.379 (global)** | REF | **0.379 (global)** |
| Water | 30.00 (3.40 −264.50) | 0.002 | 0.60 (0.14 −2.68) | 0.508 | 1.85 (0.51 −6.66) | 0.347 | 1.85 (0.51 −6.66) | 0.347 |
| Wildlife | 9.82 (1.25 −77.20) | 0.030 | 0.61 (0.21 −1.75) | 0.357 | 0.73 (0.28 −1.91) | 0.518 | 0.73 (0.28 −1.91) | 0.518 |
| Other environmental[d] | 8.00 (1.01 −63.23) | 0.049 | 0.77 (0.28 −2.15) | 0.624 | 0.91 (0.36 −2.32) | 0.847 | 0.91 (0.36 −2.32) | 0.847 |

| | IncFIB(AP001918)[a] | | IncI1(1-alpha)[b,e] | | IncFII[a] | |
|---|---|---|---|---|---|---|
| **Source type** | **OR (95% CI)** | ***p*-value** | **OR (95%CI)** | ***p*-value** | **OR (95% CI)** | ***p*-value** |
| Swine manure pit | REF | **0.186 (global)** | REF | **0.594 (global)** | REF | **0.425 (global)** |
| Water | 2.80 (0.83 −9.49) | 0.097 | 0.23 (0.01 −4.40) | 0.331 | 0.86 (0.21 −3.41) | 0.827 |
| Wildlife | 2.39 (0.91 −6.26) | 0.076 | 1.39 (0.28 −7.00) | 0.689 | 0.42 (0.14 −1.29) | 0.130 |
| Other environmental[d] | 2.63 (1.01 −6.85) | 0.047 | 1.09 (0.22 −5.39) | 0.917 | 0.52 (0.18 −1.52) | 0.232 |

[a] The random intercept to account for clustering by site or animal was not retained in the model, thus ordinary logistic regression was used.

[b] Included a random intercept for clustering by site. Variance components were: *tet(B)* 0.10 (95%CI: 0.00−4.12); *bla*TEM-1 0.24 (95%CI: 0.03−1.80); *sul1* 0.27 (95%CI: 0.02−3.12); *aph(3")-Ib* 0.05 (95%CI: 0.00−28.92); *aph(6)-Id* 0.05 (95%CI: 0.00−28.92); IncI1(1-alpha) 0.93 (95%CI: 0.13−6.49).

[c] Adjusted for confounding by year of sampling

[d] Includes soil and dumpster isolates.

[e] A random intercept to account for clustering of isolates obtained from the same animal/dumpster/manure pit was retained in this model (variance components 0.54, 95%CI: 0.00−301.89).

**Table 6. Contrasts from logistic regression models[a,b,c] (Table 5) assessing the statistically significant associations between source type and the occurrence of select antimicrobial resistance genes in phenotypically resistant *Escherichia coli* isolates collected from wildlife, swine manure pits, and environmental sources in southern Ontario, 2011−2013 (n = 200, dataset A).**

| | *tet(B)*[a,b] | | *bla*$_{TEM-1}$[b] | | *sul2*[c] | |
|---|---|---|---|---|---|---|
| **Contrast** | **OR (95%CI)** | ***p*-value** | **OR (95%CI)** | ***p*-value** | **OR (95%CI)** | ***p*-value** |
| Water vs. wildlife | 3.23 (0.99 −10.52) | 0.051 | 1.91 (0.59 −6.16) | 0.280 | 3.05 (1.09 −8.52) | 0.033 |
| Other environmental[d] vs. wildlife | 1.12 (0.51−2.48) | 0.772 | 0.42 (0.19 −0.90) | 0.025 | 0.81 (0.38 −1.75) | 0.600 |
| Other environmental[d] vs. water | 0.35 (0.11−1.09) | 0.071 | 0.22 (0.06 −0.73) | 0.014 | 0.27 (0.09 −0.75) | 0.012 |

[a] Adjusted for confounding by year of sampling.

[b] Site of sampling was retained as a random intercept.

[c] The random intercept to account for clustering by site or animal was not retained in the model, thus ordinary logistic regression was used.

[d] Includes soil and dumpster isolates.

more likely to carry organisms displaying resistance [23, 27, 39–42]. It is apparent that regional approaches are most useful in understanding complex ecological issues such as the movement of AMR between different sources. Selection of isolates for this study was based on previously demonstrated phenotypic resistance, and, although biased, this work provides insights about the nature of AMR transmission between different wildlife, livestock, and environmental sources in the Grand River watershed. The majority of *E. coli* isolates in this study originated from raccoons and soil (~70%), followed by swine manure pits and water isolates, with very few isolates from additional sources (i.e., dumpsters, skunks, opossums). In part, this distribution was related to sampling methods of previous work (e.g., fewer samples were obtained from swine manure pits than from raccoons; see Bondo *et al.* [19]). Interestingly, the number of resistant *E. coli* obtained from wildlife and soil samples in the previous wildlife study was nearly identical between conservation areas (n = 74) and swine farms (n = 72), and the overall prevalence of resistant *E. coli* among raccoon fecal samples did not significantly differ by location type [19]. With location type as a proxy for the impact of the local environment (i.e., farm vs. conservation area) on the occurrence of AMR determinants in wildlife isolates, it appeared that the presence of AMR determinants in general does not vary within this region, regardless of proximity to agriculture. Our present analysis of population structure suggests that mixing of isolates from different sources and location types is frequently occurring, with no obvious clustering of *E. coli* cgMLST subtypes by either of these factors, or by the presence of AMR. However, along with previous findings of distinct resistance patterns and AMR genes between swine farms and conservation areas [19], our examination of resistance genes, predicted plasmids, and internationally important sequence types offers a more nuanced picture of the distribution and potential movement of AMR within this study population.

A diversity of AMR genes and plasmid Inc types were identified in our study, some of which varied significantly depending on the sampling source and location type. Although most genes and plasmid Inc groups could not be modeled due to their low overall prevalence (<5%), several of the more prevalent resistant genes (e.g., *tet(A)*, *tet(B)*, *bla*$_{TEM-1}$) and Inc groups (e.g., IncFIB[AP001918]) occurred in at least 30% of all isolates. Our results provide evidence of the complexity of AMR movement and transmission, in that the factors influencing the occurrence of each resistance determinant were variable. Several overarching patterns emerged, however. For instance, the presence of resistance genes *sul2*, and *bla*$_{TEM-1}$ were

**Table 7. Logistic regression models[a,b,c] assessing the association between source type, location type, and the occurrence of select plasmid incompatibility types and antimicrobial resistance genes in phenotypically resistant *Escherichia coli* isolates from wildlife and soil samples collected on swine farms and conservation areas in southern Ontario, 2011–2013 (n = 146, dataset B).**

| Independent variable | Sub-category | *tet(A)* | | *tet(B)* | | *bla*TEM-1 | |
|---|---|---|---|---|---|---|---|
| | | OR (95% CI) | *p*-value | OR (95% CI) | *p*-value | OR (95% CI) | *p*-value |
| Location type | Conservation area | REF | 0.031[a] (global) | REF | 0.331[b] (global) | REF | 0.437[b] (global) |
| | Swine farm | 2.06 (1.06–4.01) | 0.033 | 1.64 (0.60–4.44) | 0.331 | 0.69 (0.26–1.77) | 0.438 |
| Source type | Soil | REF | 0.698[b] (global) | REF | 0.938[b] (global) | REF | 0.029[b,c] (global) |
| | Wildlife | 0.88 (0.44–1.76) | 0.716 | 0.97 (0.45–2.08) | 0.938 | 3.13 (1.35–7.22) | 0.008 |

| | | *sul1* | | *sul2* | | *ant(3'')-Ia* | |
|---|---|---|---|---|---|---|---|
| | | OR (95% CI) | *p*-value | OR (95% CI) | *p*-value | OR (95% CI) | *p*-value |
| Location type | Conservation area | REF | 0.998[a] (global) | REF | 0.914[a] (global) | REF | 0.333[a] (global) |
| | Swine farm | 1.0 (0.29–3.50) | 0.998 | 0.96 (0.44–2.08) | 0.914 | 1.54 (0.64–3.75) | 0.336 |
| Source type | Soil | REF | 0.151[a] (global) | REF | 0.553[a] (global) | REF | 0.655[a] (global) |
| | Wildlife | 2.07 (0.77–5.57) | 0.151 | 1.27 (0.58–2.76) | 0.553 | 0.82 (0.34–1.97) | 0.655 |

| | | *aph(3'')-Ib* | | *aph(6)-Id* | | IncFIB(AP001918) | |
|---|---|---|---|---|---|---|---|
| | | OR (95% CI) | *p*-value | OR (95% CI) | *p*-value | OR (95% CI) | *p*-value |
| Location type | Conservation area | REF | 0.011[a] (global) | REF | 0.011[a] (global) | REF | 0.031[a] (global) |
| | Swine farm | 2.45 (1.22–4.92) | 0.012 | 2.45 (1.22–4.92) | 0.012 | 2.07 (1.06–4.04) | 0.032 |
| Source type | Soil | REF | 0.492[b] (global) | REF | 0.492[b] (global) | REF | 0.738[a] (global) |
| | Wildlife | 0.78 (0.38–1.60) | 0.492 | 0.78 (0.38–1.60) | 0.492 | 0.89 (0.46–1.72) | 0.738 |

| | | IncI1(1-alpha) | | IncFII | | | |
|---|---|---|---|---|---|---|---|
| | | OR (95% CI) | *p*-value | OR (95% CI) | *p*-value | | |
| Location type | Conservation area | REF | 0.203[b] (global) | REF | 0.571[b] (global) | | |
| | Swine farm | 0.39 (0.09–1.66) | 0.203 | 1.33 (0.49–3.59) | 0.573 | | |
| Source type | Soil | REF | 0.516[b] (global) | REF | 0.614[a] (global) | | |
| | Wildlife | 1.36 (0.54–3.43) | 0.516 | 0.77 (0.29–2.09) | 0.615 | | |

[a] The random intercept for clustering by site or animal was not retained in the model, thus ordinary logistic regression was used.

[b] Included a random intercept for clustering at the site-level. Variance components were: *tet(A)* source type 0.22 (95%CI: 0.03–1.72); *tet(B)* source type 0.13 (95%CI: 0.00–4.63); *tet(B)* location type 0.16 (95%CI: 0.00–2.91); *bla*TEM-1 location type 0.22 (95%CI: 0.02–2.33); *bla*TEM-1 source type 0.24 (95% CI: 0.02–2.40); *sul1* location type 0.40 (95%CI: 0.05–3.39); *sul1* source type 0.41 (95%CI: 0.05–3.36); *aph(3'')-Ib* source type 0.32 (95%CI: 0.02–3.50); *aph(6)-Id* source type 0.32 (95%CI: 0.02–3.50); IncI1(1-alpha) source type 0.73 (95%CI: 0.11–4.69); IncI1(1-alpha) location type 0.63 (95%CI: 0.10–3.95).

[c] Adjusted for confounding by year of sampling.

consistently higher in water-derived isolates compared to all other sources, possibly related to upstream wastewater treatment plants along the Grand River [43]. Conversely, for resistance determinants that were analyzed by location type (i.e., wildlife and soil isolates from swine farms or conservation areas), the odds of identifying many of these genes and predicted plasmids were significantly greater in swine farm environments compared to conservation areas, regardless of sampling source, which suggests that agriculture may be the primary source of these particular AMR genes and plasmids for wildlife in our study. Combining findings from the overall dataset with the subset analysis of wildlife and soil isolates provides some additional clues about potential primary sources and the movement of certain resistance genes. Tetracycline genes *tet(A)* and *tet(B)*, for instance, demonstrated contrasting epidemiological patterns; the odds of an isolate carrying *tet(A)* were significantly greater on swine farms than in conservation areas for the wildlife and soil isolates examined, but this gene was not associated with swine manure pits, or any other particular source overall. By contrast, the odds of identifying *tet(B)* were no different between swine farms and conservation areas, but the odds of this gene were significantly greater in water and swine manure isolates compared to those from wildlife,

and other environmental samples. Thus, for certain resistance determinants such as *tet(A)*, it appears that the local environment may act as an important predictor of their distribution in wildlife. Conversely, the distribution of other genes such as *tet(B)* appears to be vary with the sampling source, rather than the local environment. However, keeping in mind the extensive nature of this particular watershed and the convergence of many different waterways, the difference between farm environments and conservation areas in this region that share broadly similar geographic characteristics may not, in fact, be sufficiently different to influence the distribution of certain genes and plasmids (e.g., *tet(B)*).

A strikingly similar pattern of these two tetracycline resistance genes was also documented in wild small mammals (i.e., mice, voles, shrews) captured in the same study region in 2008; *tet(A)*–but not *tet(B)*–was significantly more likely to be identified in animals captured on swine farms compared to residential areas [39]. When we previously examined the epidemiology of these two genes among *E. coli* in the swine farm environment in detail, no clear patterns emerged by farm location, year of sampling, or source (raccoons, soil, swine manure pits) [28]. Antimicrobial use data for these swine farms were unavailable, but tetracyclines are often used in the Canadian swine industry [44]. The impact of naturally occurring tetracycline resistance in the soil is unknown in this context, and merits further consideration in future work examining the distribution of AMR in wildlife in different types of environments [45].

The only resistance determinant that was consistently associated with source type for both the overall dataset, and for the wildlife subset, was $bla_{TEM-1}$; moderate clustering by sampling site was also consistently noted for this gene (for both datasets). With the odds of identifying $bla_{TEM-1}$ being significantly higher in water and wildlife, and with clustering by site regardless of location type, this suggests an anthropogenic, wildlife, or environmental source besides the swine farms sampled in this study. In the future, multivariable modelling, along with larger sample sizes, particularly for certain sources that were under-sampled in our study (e.g., dumpsters), may help clarify the importance of these sources, and other wildlife, in the occurrence and distribution of AMR genes. Although year of sampling was not consistently a confounder in our models, this factor merits consideration in future work, as the patterns and distributions of resistance genes might be expected to gradually change over longer periods of time, and this possibility should not be ruled out based on lack of significance in our three-year study.

In addition to a handful of serovars representing potentially pathogenic serotypes (e.g., O103), *E. coli* isolates representing major sequence types commonly associated with human bloodstream and urinary tract infections were also identified in raccoon, water, and soil isolates (e.g., ST69, ST131, ST95) [37], and many of these isolates also displayed MDR. The vast majority of these particular raccoon and soil isolates were identified in conservation area 1 (n = 4/6 isolates), despite uniform sampling of raccoons and soil across 10 sampling sites in the previous wildlife study [19]. Of particular note, a ciprofloxacin-resistant *E. coli* ST131 isolate was identified in a raccoon from conservation area 1; fluoroquinolone-resistant *E. coli* ST131 are widely recognized as a major cause of community and hospital-acquired urinary tract infections in humans across the world [46]. Apart from typical outdoor recreational activities enjoyed by visitors to conservation area 1, there was no apparent source of contamination (e.g., sewage, landfills) in the general vicinity which would account for these findings of major sequence types concentrated at this particular site. Although these isolates could not be analyzed statistically due to their low overall prevalence, their appearance exclusively in water sources and conservation areas suggests that agriculture is not a major risk factor for these strains. That said, indirect exposure to agricultural sources via water run-off cannot be ruled out, given the widespread presence of agriculture in this watershed, and the numerous waterways which feed into water sources within these conservation areas. Our findings of human-

associated sequence types, AmpC-producers, and genes conferring resistance to quinolones among raccoon and soil samples strictly from conservation areas, are consistent with previous epidemiological findings by Bondo *et al.* [19], who demonstrated that resistant *E. coli* were significantly more likely to be detected on the paws of raccoons in conservation areas compared to swine farms, and suggested this might be related to raccoons foraging human garbage in those locations. Our findings are also in line with recent work by Worsley-Tonks *et al.* [23] which provides further evidence that proximity to anthropogenic sources can influence the carriage of AMR by raccoons. In their Chicago-based study, urban raccoons trapped near wastewater treatment plants had a greater likelihood of carrying plasmid-associated resistance genes compared to those sampled on sites without such sources of AMR contamination nearby [23]. Related work comparing *E. coli* in raccoons and domestic dogs in urban parks in Chicago revealed that exchange of extended-spectrum cephalosporin-resistant *E. coli* may be occurring between these two species (based on a single-nucleotide polymorphism analysis), but dogs are not likely to be a major source of these organisms for urban raccoons, since the prevalence of these resistant *E. coli* was significantly lower in dogs compared to raccoons (16.5% vs. 56.9%, respectively) [24]. Given the opportunistic foraging nature of raccoons, numerous routes of transmission may plausibly contribute to the acquisition of AMR determinants by these animals from human sources in conservation areas, including human refuse, dog refuse, littered garbage, or dumpsters [19].

## Limitations

The selection of isolates based on demonstrated phenotypic resistance contributes to a biased representation of certain outcomes; for instance, the prevalence of pathogenic *E. coli* among the isolates included in our study should not necessarily be considered as representative of the population of untyped *E. coli* obtained in the previous wildlife study [19]. Any outcomes presented in this work must therefore be interpreted as being within the population of resistant *E. coli* from these sampling sources. Knowing that the prevalence of resistant *E. coli* varied considerably between different sources (57% of swine manure isolates; 22% of water isolates; 14% of dumpster isolates; 6–7% of raccoon paw and fecal isolates; 10% of soil isolates), the measures of association reported within our study should not be interpreted literally as the precise odds of identifying a certain gene or plasmid in one source compared to another. Instead, our analyses are targeted at addressing where certain AMR determinants are most concentrated within this population of resistant *E. coli*, in order to postulate primary sources, and to identify evidence of potential transmission between sources. Using cross-sectional data to address these research questions has inherent limitations, particularly for determining transmission, and, by extension, the direction of transfer.

A number of methodological limitations also merit consideration. The prediction of AMR genes using *in silico* tools generally appears to be highly sensitive and specific, similar to other work [47, 48] and our previous assessment [28]. Comparison of the performance of *in silico* tools between different studies should be done cautiously, however, since test performance depends on more than the particular *in silico* AMR identification tool used: the identity and coverage cut-offs used (where applicable), the quality of sequencing data, and other aspects of bioinformatics pipelines (e.g., quality control) are all potential factors which can impact test results. In addition, we did not assess AMR prediction of a major drug class (quinolones), since the focus on this work was acquired resistance genes, and not chromosomal resistance mechanisms. Although all MDR isolates were positive for plasmid Inc groups that have previously been associated with the resistance genes identified, further confirmation that these genes were in fact contained within the plasmid rather than on the chromosome, is needed.

Without the use of selective media, certain pathogenic *E. coli* serovars identified here may be underrepresented (i.e., non-O157 verotoxigenic *E. coli*; VTEC), and the presumed virulence of potentially pathogenic strains (based on serovar) requires further confirmation in future investigations to facilitate comparison to other recent surveys of raccoons that have demonstrated these animals may shed VTEC and enteropathogenic *E. coli* (EPEC) in their feces [25].

Finally, the inclusion of location type as a predictor in our hypothesis-generating study may need to be revisited in future work, since the characterization of agricultural sources in the ecosystem is undoubtedly more complex than what we have suggested here, and the contribution of upstream agricultural sources that were not directly sampled within our study were not accounted for in our analyses. Consequently, a lack of association with location type in our study should therefore not be taken to mean that agriculture is not an important risk factor for certain resistance genes and plasmids in this region.

## Conclusions

Recognizing that regional approaches are needed to better understand complex issues such as AMR which affects all components of the ecosystem, our cross-sectional study addresses knowledge gaps regarding the distribution and potential transmission of resistance genes and predicted plasmids in a southern Ontario ecosystem. Using isolates obtained from wildlife, swine manure pits, and environmental sources, this work contributes to a more comprehensive examination of the role of wildlife in the maintenance and transmission of AMR determinants within the Grand River watershed. While some resistance genes were associated with certain sources (i.e., $bla_{\text{TEM-1}}$ in water and wildlife), others were associated with certain location types (i.e., *aph(3")-Ib*, *tet(A)*, and *aph(6)-Id* were higher on swine farms than conservation areas). Meanwhile, major sequence types frequently implicated in human illness (i.e., ST69, ST131, ST95) were found exclusively in isolates from water, and in raccoon and soil-derived isolates on conservation areas, but not on swine farms. In combination with previous work on this raccoon population demonstrating that AmpC-producing *E. coli* were almost exclusively identified in conservation areas, these findings are suggestive of anthropogenic sources for these particular types of resistance determinants. Overall, the variability in the distribution of different cgMLST subtypes, sequence types, genes, and plasmid Inc groups by source and location type underscores the complex set of factors and interactions which can influence the distribution of various determinants of resistance. Based on our findings, it is clear that apparently healthy wildlife may act as sentinels and sources of AMR contamination and potentially pathogenic *E. coli* for humans, and that some may differ in their carriage of certain sequence types, genes, and plasmids according to their local environment. Future investigations focused on intervention-based approaches, integration of antimicrobial use data on farms, sampling of additional livestock sources, and use of whole-genome sequence data to confirm plasmid structure and associated genes would help to address certain major knowledge gaps in this field.

## Supporting information

**S1 File. Gene accession numbers for *Escherichia coli* isolates from previous wildlife study.** (XLSX)

**S1 Table. Multi-locus sequence types identified using whole-genome sequence data of phenotypically resistant *Escherichia coli* isolates obtained from wildlife, swine manure pits, and environmental sources in southern Ontario, Canada, 2011–2013.** (DOCX)

**S2 Table. Serotypes identified using whole-genome sequence data of phenotypically resistant *Escherichia coli* isolates obtained from wildlife, swine manure pits, and environmental sources in southern Ontario, Canada, 2011–2013.**
(DOCX)

**S3 Table. Distribution of plasmid incompatibility types identified using whole-genome sequence data by source type for phenotypically resistant *Escherichia coli* isolates from wildlife, swine manure pits, and environmental sources in southern Ontario, Canada, 2011–2013.**
(DOCX)

## Acknowledgments

Erin Harkness, Samantha Kagan, and Mary Thompson assisted with the collection of field data. Bryan Bloomfield engaged the participation of landowners, and Tami Harvey and Barbara Jefferson submitted lab samples. Culture of samples, and submission of isolates for sequencing was performed and coordinated by the McEwen Lab at the Canadian Research Institute for Food Safety (Sarah Martz), University of Guelph, Public Health Agency of Canada (Andrea Nesbitt) and the National Microbiology Laboratory at Guelph, Public Health Agency of Canada (Antimicrobial Resistance Reference Laboratory, Andrea Desruisseau, Chad Gill). Ketna Mistry performed the sequencing of *E. coli* isolates (National Microbiology Laboratory at Guelph, Public Health Agency of Canada, OIE Reference Lab for Salmonellosis). Gabhan Chalmers performed DNA extractions that were conducted in the Boerlin Laboratory at the University of Guelph. Laura Mataseje assisted with acquisition of sequence data. Water sample collection and *E. coli* isolation from these samples was performed by the Ontario Ministry of the Environment, Conservation and Parks.

## Author Contributions

**Conceptualization:** Nadine A. Vogt, Benjamin M. Hetman, David L. Pearl, Richard J. Reid-Smith, E. Jane Parmley, Michael R. Mulvey, Claire M. Jardine.

**Data curation:** Nadine A. Vogt, Benjamin M. Hetman, Adam A. Vogt, Stefanie Kadykalo, Kim Ziebell, Kristin J. Bondo.

**Formal analysis:** Nadine A. Vogt, Benjamin M. Hetman, Adam A. Vogt, David L. Pearl.

**Funding acquisition:** David L. Pearl, Richard J. Reid-Smith, E. Jane Parmley, Michael R. Mulvey, Claire M. Jardine.

**Investigation:** Nadine A. Vogt, Stefanie Kadykalo, Kim Ziebell, Amrita Bharat, Nicol Janecko, Samantha E. Allen, Kristin J. Bondo.

**Methodology:** Nadine A. Vogt, Benjamin M. Hetman.

**Project administration:** Nadine A. Vogt, Richard J. Reid-Smith, E. Jane Parmley, Michael R. Mulvey, Claire M. Jardine.

**Resources:** Nadine A. Vogt, Benjamin M. Hetman, Adam A. Vogt, Claire M. Jardine.

**Software:** Nadine A. Vogt, Benjamin M. Hetman, Adam A. Vogt.

**Supervision:** David L. Pearl, Richard J. Reid-Smith, E. Jane Parmley, Michael R. Mulvey, Nicole Ricker, Claire M. Jardine.

**Validation:** Nadine A. Vogt, Benjamin M. Hetman, Adam A. Vogt, David L. Pearl.

**Visualization:** Nadine A. Vogt, Benjamin M. Hetman.

**Writing – original draft:** Nadine A. Vogt.

**Writing – review & editing:** Nadine A. Vogt, Benjamin M. Hetman, Adam A. Vogt, David L. Pearl, Richard J. Reid-Smith, E. Jane Parmley, Stefanie Kadykalo, Kim Ziebell, Amrita Bharat, Michael R. Mulvey, Nicol Janecko, Nicole Ricker, Samantha E. Allen, Kristin J. Bondo, Claire M. Jardine.

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
