## [Decision Letter · Decision Letter 0]

16 Feb 2022

PONE-D-21-40450Using whole-genome sequence data to examine the epidemiology of antimicrobial resistance in Escherichia coli from wild meso-mammals and environmental sources on swine farms, conservation areas, and the Grand River watershed in southern OntarioPLOS ONE

Dear Dr. Vogt,

Thank you for submitting your manuscript to PLOS ONE. After careful consideration, we feel that it has merit but does not fully meet PLOS ONE’s publication criteria as it currently stands. Therefore, we invite you to submit a revised version of the manuscript that addresses the points raised during the review process.

Please address all questions raised and provide clarifications based on reviewer comments.

We look forward to receiving your revised manuscript.

Kind regards,

Iddya Karunasagar

Academic Editor

PLOS ONE

Journal Requirements:

Additional Editor Comments:

The reviewers have pointed out number of aspects of the manuscript that need clarifications and revision. Please address all comments point by point.

Reviewers' comments:

Reviewer's Responses to Questions

**Comments to the Author**

1. Is the manuscript technically sound, and do the data support the conclusions?

Reviewer #1: Partly

Reviewer #2: Yes

2. Has the statistical analysis been performed appropriately and rigorously? 

Reviewer #1: Yes

Reviewer #2: Yes

3. Have the authors made all data underlying the findings in their manuscript fully available?

Reviewer #1: Yes

Reviewer #2: Yes

4. Is the manuscript presented in an intelligible fashion and written in standard English?

Reviewer #1: Yes

Reviewer #2: Yes

5. Review Comments to the Author

Reviewer #1: Using whole-genome sequence data to examine the epidemiology of antimicrobial resistance in Escherichia coli from wild meso-mammals and environmental sources on swine farms, conservation areas, and the Grand River watershed in southern Ontario

Manuscript Number: PONE-D-21-40450

The study derived the whole-genome sequence data of Escherichia coli isolates with demonstrated antibiotic resistance phenotypes. The isolates belonged to a previous wildlife study and a public health surveillance program. In total, 200 isolates from water, wildlife, swine manure pit, and other environmental samples such as soil and dumpsters were used. The AMR genes corresponding to antibiotic resistance phenotypes were identified insilico. Plasmid incompatibility groups were also identified, more frequently among swine isolates. Although the study does not precisely identify the sources of AMR strains, certain conclusions were drawn on the frequency of occurrence of AMR strains and the genotypes, likely transfer of resistance or the acquisition of AMR genes by E. coli isolates associated with small mammals.

1. Comments:

The study relies heavily on statistical analysis for arriving at definite conclusions at the end. The MDR isolates belonging to sequence types of public health significance were isolated from a limited number of small mammals. The isolates were collected during 2011-13. Wouldn’t the resistance patterns or the distribution of E. coli have changed in these regions over the years? This is also obvious from the relatively low prevalence of MDR phenotypes and the distribution of plasmid-borne genes (e.g. β-lactamase genes) among the isolates.

2. The procedures for sample collection, treatment and isolation E. coli have already been published for these isolates. Therefore, this section can be significantly shortened (L133-161). Similarly, I guess L175-194 also refer to the procedures followed in 2011.

3. The study used whole genome sequence data of a large number of isolates (n=200). However, the distribution of isolates from different sources vary widely, with most of the isolates belonging to raccoons and soil. This might have impacted the statistical analysis, since the distribution or the odds of finding a resistance gene can vary widely when the number of isolates is not uniform.

4. The rate and frequency of isolation of pathogenic E. coli (e.g. STEC) strains depend on the special selective enrichment regimes used for their isolation. Since no such protocols have been used for the isolation of human pathogenic E. coli groups in this study, the identified serogroups might represent only a proportion of such isolates from different sources.

5. Further, there other sources of pathogenic E. coli strains such as the feces of other small animals, birds etc. Specific E. coli pathogroups are also found associated with swine. Therefore, E. coli isolates from the paw of raccoons may not simply represent “contaminants form external sources such as water or anthropogenic sources”.

6. L298-300: E. coli serovars (including O103) can often be associated with human, animal or avian sources. I agree with the authors that this serotype has caused human infections, but isn’t there a possibility that such strains can be associated naturally with the wild animals?

7. Based on the statistical analysis, the chances of finding tet(B) were higher among swine manure isolates, those of blaTEM-1 and sul2I genes were higher among isolates from water. Are there any specific reasons for this association? Are certain antibiotics employed in these swine farms?

8. In L491 it is said that tet(A) is not associated with any particular source overall. However, the following paragraph (L495-495) suggests that the tet(A) genes might have originated from the swine farms. Similarly, since tet(B) gene is more prevalent in water isolates, water was presumed to be the source of this gene in all other environments. These are contradictory. When this gene is uniformly found across different environments, how is its source and mode of dissemination predicted? Studies have shown tet genes in bacteria from pristine areas that were never exposed to antibiotics.

9. In L589-591, it is stated that small animals are at higher risk of acquiring AMR E. coil through anthropogenic contamination. L80-81 suggest small mammals could be a potential source of AMR to humans and domestic animals.

10. Lines 532-533: “Resistant E. coli were significantly more likely to be detected on the paws of raccoons in conservation areas compared to swine farms”. Why is it so?

11. The manuscript emphasizes on pathogenic and resistant E. coli strains being found on the paws of raccoons. However, there is always a possibility of such strains being associated with the gastrointestinal tracts of healthy raccoons. A recent study [ Orden et al., 2020. Raccoons (Procyon lotor) in the Madrid region of Spain are carriers of antimicrobial-resistant Escherichia coli and enteropathogenic E. coli. Zoonoses Public Health. 2021; 68: 69– 78] have reported the presence of pathogenic and AMR E. coli in the feces of small mammals like raccoons and suggest these as the sources of such strains to humans. Certain serovars (e.g. O77, O25) were found both on raccoon paws and faeces in your study too.

Reviewer #2: The work presented in this article contributes to a comprehensive examination of the role of wild life in the maintenance and dissemination of AMR determinants within the Grand River watershed. The work is particularly important to identify the circulation of antimicrobial resistance genes and the mobile genetic elements that facilitate their transmission among different sources including humans.

Authors could successfully relate the association of some of the antimicrobial resistance determinants to a particular environment or location types. However, the authors have not discussed or analyzed the association of phenotypic resistance and the occurrence of corresponding drug resistance genes among different sources.

Comments

1. Authors have used a significant amount of data from their previous study, this causes repetition of data.

2. Number of samples used from each source is not clearly mentioned in the methodology

6. PLOS authors have the option to publish the peer review history of their article (what does this mean?). If published, this will include your full peer review and any attached files.

Reviewer #1: No

Reviewer #2: No

---

## [Author Response · Author response to Decision Letter 0]

23 Feb 2022

We thank reviewers for their thoughtful comments and suggestions that have helped to improve the manuscript. Our comments are below. Lines numbers correspond to the tracked changes version. 

Reviewer #1: Using whole-genome sequence data to examine the epidemiology of antimicrobial resistance in Escherichia coli from wild meso-mammals and environmental sources on swine farms, conservation areas, and the Grand River watershed in southern Ontario

Manuscript Number: PONE-D-21-40450

The study derived the whole-genome sequence data of Escherichia coli isolates with demonstrated antibiotic resistance phenotypes. The isolates belonged to a previous wildlife study and a public health surveillance program. In total, 200 isolates from water, wildlife, swine manure pit, and other environmental samples such as soil and dumpsters were used. The AMR genes corresponding to antibiotic resistance phenotypes were identified in silico. Plasmid incompatibility groups were also identified, more frequently among swine isolates. Although the study does not precisely identify the sources of AMR strains, certain conclusions were drawn on the frequency of occurrence of AMR strains and the genotypes, likely transfer of resistance or the acquisition of AMR genes by E. coli isolates associated with small mammals.

1. Comments:

The study relies heavily on statistical analysis for arriving at definite conclusions at the end. The MDR isolates belonging to sequence types of public health significance were isolated from a limited number of small mammals. The isolates were collected during 2011-13. Wouldn’t the resistance patterns or the distribution of E. coli have changed in these regions over the years? This is also obvious from the relatively low prevalence of MDR phenotypes and the distribution of plasmid-borne genes (e.g. β-lactamase genes) among the isolates.

Thank you for the feedback. It's true that we lean heavily on our statistical analysis for interpretation, since we used an epidemiological approach to explore the frequencies, patterns and distributions of resistance genes and plasmid replicons in different sources, and location types. We agree that the resistance patterns or distribution of E. coli may have changed in these regions over the years, thus, we opted to account for year as a potential confounder in our analyses (see Figure 2). We've added a line in the discussion to convey the point you make (lines 610-613). 

2. The procedures for sample collection, treatment and isolation E. coli have already been published for these isolates. Therefore, this section can be significantly shortened (L133-161). Similarly, I guess L175-194 also refer to the procedures followed in 2011.

Thank you for pointing this out, these sections have been shortened, and only the most important details have been retained (lines 155-211).

3. The study used whole genome sequence data of a large number of isolates (n=200). However, the distribution of isolates from different sources vary widely, with most of the isolates belonging to raccoons and soil. This might have impacted the statistical analysis, since the distribution or the odds of finding a resistance gene can vary widely when the number of isolates is not uniform.

It's true that unbalanced data can be problematic and cause modeling issues. In our analyses, we opted to reclassify sources into larger groups (Fig 1) to ensure adequate statistical power, and more balanced sampling of categories. We ended up with the distribution shown in Table 2 for our source type analyses: wildlife (n=73), swine manure pit (n=31), water (n=20), and "other environmental" (n=76). For the subset analysis on swine farms and conservation areas, the source distribution ended up being even: wildlife (n=73), soil (n=73). We report all 95% confidence intervals around our OR estimates to ensure it's clear to the reader that certain categories which are smaller have wider confidence intervals. Although the source distributions for the broad analysis are not ideal, our study represents a hypothesis-generating study, and we were opportunistic with available data from available sources, and careful to focus only on one variable at a time, either "location type" or "source type" (while accounting for potential confounding effects of sampling year). In line with the suggestions above, we have suggested larger sample sizes of certain source categories are needed in future work (lines 607-610). 

4. The rate and frequency of isolation of pathogenic E. coli (e.g. STEC) strains depend on the special selective enrichment regimes used for their isolation. Since no such protocols have been used for the isolation of human pathogenic E. coli groups in this study, the identified serogroups might represent only a proportion of such isolates from different sources.

Thank you for pointing out this limitation. It has been added to the manuscript (lines 680-682). 

5. Further, there other sources of pathogenic E. coli strains such as the feces of other small animals, birds etc. Specific E. coli pathogroups are also found associated with swine. Therefore, E. coli isolates from the paw of raccoons may not simply represent “contaminants form external sources such as water or anthropogenic sources”.

We have ensured that we have qualified our discussions of pathogenic E. coli strains with the word "potential" (lines 713-714; line 682) and maintain that our findings of certain strains nearly exclusively in conservation areas (area 1 in particular) are "suggestive" of anthropogenic exposures (line 708), which doesn't exclude other possible sources. 

6. L298-300: E. coli serovars (including O103) can often be associated with human, animal or avian sources. I agree with the authors that this serotype has caused human infections, but isn’t there a possibility that such strains can be associated naturally with the wild animals?

We have revised the results and discussion to reflect the uncertainty of the pathogenicity of these serovars (lines 354-358; lines 614-615; lines 682-683). 

7. Based on the statistical analysis, the chances of finding tet(B) were higher among swine manure isolates, those of blaTEM-1 and sul2I genes were higher among isolates from water. Are there any specific reasons for this association? Are certain antibiotics employed in these swine farms?

Good questions. Unfortunately, we didn't have access to antimicrobial use data on the swine farms, and have suggested it for future work (lines 583-585). We have provided a possible explanation for the higher prevalence of blaTEM-1 and sul2 genes in water (lines 555-556). 

8. In L491 it is said that tet(A) is not associated with any particular source overall. However, the following paragraph (L495-495) suggests that the tet(A) genes might have originated from the swine farms. Similarly, since tet(B) gene is more prevalent in water isolates, water was presumed to be the source of this gene in all other environments. These are contradictory. When this gene is uniformly found across different environments, how is its source and mode of dissemination predicted? Studies have shown tet genes in bacteria from pristine areas that were never exposed to antibiotics.

Thank you for catching this error in over-interpretation of our analyses (we will need multivariable models to assess location type and source type in the same model). We inferred that primary sources of certain genes/plasmids would have a higher prevalence (this is an inference however). Uniform distributions don't tell us a whole lot, other than widespread dissemination, and that no particular source should necessarily be suspected as a primary source. We tried to simultaneously combine the broad scale analysis with the location type analysis, but this is over-interpretation, and beyond the ability of our hypothesis-generating study. We have removed the contradictory section, and revised the interpretation, and added another reference (lines 577-587). 

9. In L589-591, it is stated that small animals are at higher risk of acquiring AMR E. coil through anthropogenic contamination. L80-81 suggest small mammals could be a potential source of AMR to humans and domestic animals.

Although these statements are not mutually exclusive, we have added several phrases to capture both concepts throughout (line 70; line 682; lines 712-713). 

10. Lines 532-533: “Resistant E. coli were significantly more likely to be detected on the paws of raccoons in conservation areas compared to swine farms”. Why is it so?

The previous authors hypothesized that this was related to foraging behaviour by raccoons in dumpsters. We have added a line to convey this hypothesis (lines 638-639). 

11. The manuscript emphasizes on pathogenic and resistant E. coli strains being found on the paws of raccoons. However, there is always a possibility of such strains being associated with the gastrointestinal tracts of healthy raccoons. A recent study [ Orden et al., 2020. Raccoons (Procyon lotor) in the Madrid region of Spain are carriers of antimicrobial-resistant Escherichia coli and enteropathogenic E. coli. Zoonoses Public Health. 2021; 68: 69– 78] have reported the presence of pathogenic and AMR E. coli in the feces of small mammals like raccoons and suggest these as the sources of such strains to humans. Certain serovars (e.g. O77, O25) were found both on raccoon paws and faeces in your study too.

Thank you for pointing out this reference, it was missed. We have added these findings to the discussion (lines 682-685), and highlighted the zoonotic concern of potentially pathogenic E. coli strains isolated from raccoons (lines 712-713). 

Reviewer #2: The work presented in this article contributes to a comprehensive examination of the role of wildlife in the maintenance and dissemination of AMR determinants within the Grand River watershed. The work is particularly important to identify the circulation of antimicrobial resistance genes and the mobile genetic elements that facilitate their transmission among different sources including humans. Authors could successfully relate the association of some of the antimicrobial resistance determinants to a particular environment or location types. However, the authors have not discussed or analyzed the association of phenotypic resistance and the occurrence of corresponding drug resistance genes among different sources.

We opted to perform only a preliminary scan of class-level sensitivity and specificity of in silico identification of AMR for our hypothesis-generating epidemiologic study. The aim of this work was focused on the epidemiology of those genes in different sources and location types, thus, we argue that a fine-scaled assessment of in silico methods was not necessary for our objective. The WGS data from the wildlife study have been deposited and the phenotypic data for those isolates was previously made available by Bondo et al. (see data URLs for those previous manuscripts), should readers want to assess gene level sensitivity and specificity for these isolates. There were many possible avenues for data analysis, and we decided to focus on areas that were supported by our expertise (i.e., epidemiology). In the interest of keeping our manuscript as streamlined as possible, and minimize the addition of further tables and analyses, we opted to keep our AMR phenotype-genotype assessment as it currently is, as a secondary objective to provide a course-grained validation of our statistical analyses. 

Comments

1. Authors have used a significant amount of data from their previous study, this causes repetition of data.

While it's true that we have used the sequenced isolates from swine farms in this manuscript, we argue these manuscripts each have a different scope and different objectives regarding the epidemiology of AMR. The previous manuscript was focused on the epidemiology of AMR in E. coli (and in Salmonella) at the swine farm level, and the analyses addressed the following predictor variables: sampling year, site id, and source type. This manuscript builds on those previous analyses and examines location type and another source (water) from a broader geographic region. The previous manuscript is focused on a small-scale, local analysis, whereas the present manuscript is focused on broader scale analyses and epidemiological patterns which may be related to different types of environments (farm vs. conservation area), and sources (i.e., water). Combining this present manuscript with our previous analyses looking at both Salmonella and E. coli would have negatively affected interpretability, presentation of the data, discussion flow, and manuscript length. The end of the introduction has been re-worked to ensure it is clear why the present work was performed, and how it differs from the previous studies (lines 97-110). 

2. Number of samples used from each source is not clearly mentioned in the methodology

The number of isolates available for sequencing or that were previously sequenced have been added to the methods section (lines 158-161; lines 210-211).

---

## [Decision Letter · Decision Letter 1]

21 Mar 2022

PONE-D-21-40450R1Using whole-genome sequence data to examine the epidemiology of antimicrobial resistance in Escherichia coli from wild meso-mammals and environmental sources on swine farms, conservation areas, and the Grand River watershed in southern OntarioPLOS ONE

Dear Dr. Vogt,

Thank you for submitting your manuscript to PLOS ONE. After careful consideration, we feel that it has merit but does not fully meet PLOS ONE’s publication criteria as it currently stands. Therefore, we invite you to submit a revised version of the manuscript that addresses the points raised during the review process.

Some minor points to be addressed as pointed out by the reviewers. 

We look forward to receiving your revised manuscript.

Kind regards,

Iddya Karunasagar

Academic Editor

PLOS ONE

Journal Requirements:

Additional Editor Comments (if provided):

The revised version looks good, but there are some minor points to be addressed. Please see the reviewer comments.

Reviewers' comments:

Reviewer's Responses to Questions

**Comments to the Author**

1. If the authors have adequately addressed your comments raised in a previous round of review and you feel that this manuscript is now acceptable for publication, you may indicate that here to bypass the “Comments to the Author” section, enter your conflict of interest statement in the “Confidential to Editor” section, and submit your "Accept" recommendation.

Reviewer #1: All comments have been addressed

Reviewer #2: All comments have been addressed

2. Is the manuscript technically sound, and do the data support the conclusions?

Reviewer #1: Yes

Reviewer #2: Yes

3. Has the statistical analysis been performed appropriately and rigorously? 

Reviewer #1: Yes

Reviewer #2: Yes

4. Have the authors made all data underlying the findings in their manuscript fully available?

Reviewer #1: (No Response)

Reviewer #2: Yes

5. Is the manuscript presented in an intelligible fashion and written in standard English?

Reviewer #1: Yes

Reviewer #2: Yes

6. Review Comments to the Author

Reviewer #1: PONE-D-21-40450R1

The manuscript is adequately revised. A few minor points need attention.

In the abstract, it is necessary to indicate that the soil isolates were 73 and dumpster isolates were 3 considering the huge difference in the number of isolates from these two sources.

Line 77: Please delete “in” after “documented”.

Line 78: Please insert a dot after [4, 17, 18]. And start with “However,”

Impact of source type and location type

An additional objective was to assess the validity of in silico identification of AMR genes, using phenotypic susceptibility test results as the gold standard. Does it mean the validity of the tool used? Does the specificity depend on the tool, targe genome or the sources?

In your study, the test sensitivity for beta lactams was 90.2% and aminoglycosides, 81.8%. Is this the same as the positive predictive value?.

Lines 161-181: The section under “Previous culture and susceptibility testing” . As I could understand, this data is from the previous study. In that case, the section can be merged with the previous section and reduced to few lines indicating only the antimicrobials tested. The methods, reference standards etc are available from the previous publication.

Table 1: Please indicate bla genes in proper format. TEM is written as a subscript of bla and in normal font (TEM is not italicized).

Reviewer #2: The authors have addressed all the reviewers comments and it can now be accepted for publication in the journal

7. PLOS authors have the option to publish the peer review history of their article (what does this mean?). If published, this will include your full peer review and any attached files.

Reviewer #1: No

Reviewer #2: No

---

## [Author Response · Author response to Decision Letter 1]

22 Mar 2022

Thank you again for your helpful feedback and insights. The lines listed here correspond to the tracked changes version. 

Reviewer #1: PONE-D-21-40450R1

The manuscript is adequately revised. A few minor points need attention.

In the abstract, it is necessary to indicate that the soil isolates were 73 and dumpster isolates were 3 considering the huge difference in the number of isolates from these two sources.

We agree it is best to report these categories separately. The change has been made. 

Line 77: Please delete “in” after “documented”.

Thank you for catching this error. 

Line 78: Please insert a dot after [4, 17, 18]. And start with “However,”

The change has been made. 

An additional objective was to assess the validity of in silico identification of AMR genes, using phenotypic susceptibility test results as the gold standard. Does it mean the validity of the tool used? Does the specificity depend on the tool, targe genome or the sources?

Interesting point. A line has been added to the discussion to elaborate on this (lines 607-611). The test performance depends largely on the AMR identification tool used, but other factors should also be considered (unlikely related to source, unless source impacts the quality of data). 

In your study, the test sensitivity for beta lactams was 90.2% and aminoglycosides, 81.8%. Is this the same as the positive predictive value?.

The test sensitivity is a measure of the ability of the test to detect true positives (true positives that test positive/all true positives), whereas the positive predictive value is a measure of the probability of a test positive result being truly positive (true positives that test positive/all test positives). While positive predictive values have been used in this context (they are typically used in clinical medicine), I would prefer to avoid them since they are heavily influenced by the overall prevalence of the “disease” or AMR gene, in this case. 

Lines 161-181: The section under “Previous culture and susceptibility testing” . As I could understand, this data is from the previous study. In that case, the section can be merged with the previous section and reduced to few lines indicating only the antimicrobials tested. The methods, reference standards etc are available from the previous publication.

This section has been shortened (lines 167-176), but important methodological information has been retained to ensure the reader can access the relevant information for the additional objective (validity of in silico AMR identification) within the same manuscript, without having to check the previous papers. 

Table 1: Please indicate bla genes in proper format. TEM is written as a subscript of bla and in normal font (TEM is not italicized).

Thank you for pointing this out. The formatting has been corrected.

---

## [Decision Letter · Decision Letter 2]

29 Mar 2022

Using whole-genome sequence data to examine the epidemiology of antimicrobial resistance in Escherichia coli from wild meso-mammals and environmental sources on swine farms, conservation areas, and the Grand River watershed in southern Ontario

PONE-D-21-40450R2

Dear Dr. Vogt,

We’re pleased to inform you that your manuscript has been judged scientifically suitable for publication and will be formally accepted for publication once it meets all outstanding technical requirements.

Kind regards,

Iddya Karunasagar

Academic Editor

PLOS ONE

Additional Editor Comments (optional):

All reviewer comments have been addressed

Reviewers' comments:

Reviewer's Responses to Questions

**Comments to the Author**

1. If the authors have adequately addressed your comments raised in a previous round of review and you feel that this manuscript is now acceptable for publication, you may indicate that here to bypass the “Comments to the Author” section, enter your conflict of interest statement in the “Confidential to Editor” section, and submit your "Accept" recommendation.

Reviewer #1: All comments have been addressed

2. Is the manuscript technically sound, and do the data support the conclusions?

Reviewer #1: Yes

3. Has the statistical analysis been performed appropriately and rigorously? 

Reviewer #1: Yes

4. Have the authors made all data underlying the findings in their manuscript fully available?

Reviewer #1: Yes

5. Is the manuscript presented in an intelligible fashion and written in standard English?

Reviewer #1: Yes

6. Review Comments to the Author

Reviewer #1: (No Response)

7. PLOS authors have the option to publish the peer review history of their article (what does this mean?). If published, this will include your full peer review and any attached files.

Reviewer #1: No

---

## [Editor Report · Acceptance letter]

31 Mar 2022

PONE-D-21-40450R2 

Using whole-genome sequence data to examine the epidemiology of antimicrobial resistance in Escherichia coli from wild meso-mammals and environmental sources on swine farms, conservation areas, and the Grand River watershed in southern Ontario 

Dear Dr. Vogt:

I'm pleased to inform you that your manuscript has been deemed suitable for publication in PLOS ONE. Congratulations! Your manuscript is now with our production department. 

Kind regards, 

on behalf of

Dr. Iddya Karunasagar 

Academic Editor

PLOS ONE